# Gastroesophageal Reflux Disease in Idiopathic Pulmonary Fibrosis: Viewer or Actor? To Treat or Not to Treat?

**DOI:** 10.3390/ph15081033

**Published:** 2022-08-22

**Authors:** Barbara Ruaro, Riccardo Pozzan, Paola Confalonieri, Stefano Tavano, Michael Hughes, Marco Matucci Cerinic, Elisa Baratella, Elisabetta Zanatta, Selene Lerda, Pietro Geri, Marco Confalonieri, Francesco Salton

**Affiliations:** 1Pulmonology Unit, Department of Medical Surgical and Health Sciences, University Hospital of Cattinara, University of Trieste, 34149 Trieste, Italy; 2Division of Musculoskeletal and Dermatological Sciences, Faculty of Biology, Medicine and Health, The University of Manchester & Salford Royal NHS Foundation Trust, Manchester M6 8HD, UK; 3Department of Experimental and Clinical Medicine, Division of Rheumatology AOUC, Scleroderma Unit, University of Florence, 50121 Florence, Italy; 4Unit of Immunology, Rheumatology, Allergy and Rare Diseases (UnIRAR), IRCCS San Raffaele Hospital, 20132 Milan, Italy; 5Department of Radiology, Cattinara Hospital, University of Trieste, 34149 Trieste, Italy; 6Department of Rheumatology, University of Padova, 35122 Padova, Italy; 724ORE Business School, Via Monte Rosa, 91, 20149 Milano, Italy

**Keywords:** idiopathic pulmonary fibrosis (IPF), gastro-oesophageal reflux disease (GERD), interstitial pneumonia, bronchoalveolar lavage fluid (BALF), interstitial lung disease (ILD), high-resolution computed tomography (HRTC)

## Abstract

Background: Idiopathic pulmonary fibrosis (IPF) is a rare and severe disease with a median survival of ∼3 years. Several risk factors have been identified, such as age, genetic predisposition, tobacco exposure, and gastro-oesophageal reflux disease (GERD). Prevalence of GERD in IPF is high and may affect 87% of patients, of whom only half (47%) report symptoms. **Objective**: The aim of this study is to review current evidence regarding the correlation between GERD and IPF and to evaluate the current studies regarding treatments for GERD-IPF. **Methods**: A review to identify research papers documenting an association between GERD and IPF was performed. **Results**: We identified several studies that have confirmed the association between GERD and IPF, with an increased acid exposure, risk of gastric aspiration and bile acids levels in these patients. Few studies focused their attention on GERD treatment, showing how antiacid therapy was not able to change IPF evolution. **Conclusions**: This review investigating the correlation between GERD and IPF has confirmed the hypothesized association. However, further large prospective studies are needed to corroborate and elucidate these findings with a focus on preventative and treatment strategies.

## 1. Introduction

Idiopathic pulmonary fibrosis (IPF) is the most common form of progressive and irreversible interstitial chronic fibroinflammatory pneumonia, accounting for approximately 50–60% of all idiopathic interstitial pneumonias (IIPs) [1,2]. The estimated incidence of IPF is 3–9 cases per 100,000 individuals in Europe and North America; and 38,000 individuals are affected by this disease every year in the USA [3,4].

The pathogenesis of IPF is highly complex and still not fully clear. IPF pathogenesis is correlated by an increase in the synthesis and release of pro-inflammatory cytokines, such as tumor necrosis factor (TNF)-α and interleukin (IL)-1, as well as some fibrous factors such as transforming growth factor (TGF)-β and platelet-derived growth factor (PDGF) [4,5,6,7,8,9,10]. Genetics and environment are potential etiologic factors, although without an established causal relationship. As for genetics, although increasing evidence is proving its role in the pathogenesis of IPF, mechanisms resulting from genetic mutations in the epithelial cells of the lung need to be better understood, which will also make therapeutic options more effective [5,6,7]. Environmental factors (e.g., cigarette smoke and air pollution) can cause an inflammatory response in the pulmonary parenchyma and consequent fibrosis, as well as gastroesophageal reflux disease (GERD)-induced inflammation, which can be considered among these factors [5,6,7,8,9,10].

IPF usually occurs in adults over 50 years old. IPF prognosis is poor, with a median survival of usually 2 to 3 years, which is comparable with that of some aggressive cancers [11,12,13,14,15]. Several factors are associated with worse evolution, such as an age greater than 70 years, a forced vital capacity (FVC) of <70% and/or a diffusing capacity of the lungs for carbon monoxide (DLCO) of <40% at diagnosis, a reduction in forced vital capacity (FVC) of ≥10% from the estimated value or a decrease in DLCO of ≥15% from the estimated value within 6-12 months of follow-up, and a decrease of >50 m in a 6-min gait test at 6 months [5,6,7,8,9,10]. Patients commonly complain about dyspnea on exertion and a nonproductive cough over several months. However, IPF (especially in the initial phases) can be relatively asymptomatic [5,6,7,8,9,10].

Traditionally, IPF treatment was based on immunosuppressants, glucocorticoids, oxygen therapy, and palliative measures. However, the PANTHER-IPF study showed that treatment with azathioprine, N-acetyl cysteine, and prednisone was associated with increased hospitalizations and mortality [5,6,7,8,9,10]. Currently, there are two drugs approved for IPF that have been demonstrated to delay lung deterioration with satisfactory safety and tolerability profiles. These two drugs are nintedanib and pirfenidone [5,6,7,8,9,10].

The purpose of this study is to review the current evidence regarding the correlation between GERD and IPF by examining a group of studies treating the topic. Recent literature was also evaluated for an update on current treatments for GERD-IPF.

## 2. Antifibrotic Drugs for IPF and Gastrointestinal Adverse Events

Currently, antifibrotic therapies (pirfenidone, nintedanib) are neither able to cure nor abort or reverse the progression of the disease, although they have proven effective in slowing down the progression, independently of age, sex, and the stage of the disease [8,9]. Benefits are in terms of a change from baseline in the FVC predicted (−3.1 and −6.3% in patients in nintedanib and comparator groups, respectively), a reduced number of exacerbations, and a bettered St. George’s Respiratory Questionnaire total score (2.92 in the nintedanib group and 4.97 in the placebo group) [10].

However, there is little information on the pharmacological interactions of these two agents in IPF patients who are usually polymedicated [10,11,12,13,14,15].

Pirfenidone belongs to the group of immunosuppressive agents. Pirfenidone (5 methyl-1-phenyl-2-[1H] pyridone) is an agent that combines anti-inflammatory and antifibrotic effects, acting on the regulation of TGF-5 activity, TNF-α and -β pathways, and cellular oxidation [5,6,7,8,9,10].

Pirfenidone reduces the accumulation of inflammatory cells and attenuates the proliferation of fibroblasts, the production of cytokines and proteins related to fibrosis, and the increased synthesis and accumulation of extracellular matrix [5,6,7,8,9,10]. Pirfenidone was approved by the US Food and Drug Administration (FDA) and the European Medicines Agency (EMA) in 2011, becoming the first drug authorized for the treatment of IPF. The most common adverse reactions are observed at the gastrointestinal, skin, and liver level. It is worth highlighting that polymedicated patients should be closely monitored because pirfenidone metabolism can be influenced in these patients by the inhibition or induction of liver enzyme systems such as cytochrome P450 1A2 (CYP1A2), CYP3A4, and P-glycoprotein (P-gp) [5,6,7,8,9,10]. The most common drugs that have demonstrated major interactions with pirfenidone are: amiodarone, enoxacin, leflunomide, fluvoxamine, aminolevulinic acid, mipomersen, mibefradil, rucaparib, and teriflunomide [5,6,7,8,9,10].

Nintedanib is an intracellular tyrosine kinase inhibitor developed for the treatment of various types of cancer (lung, ovary, renal, colorectal, and liver), as well as an antifibrotic agent [5,6,7,8,9,10]. In 2014, it was approved for the treatment of IPF in the USA and Europe and in 2019 received a new indication for systemic sclerosis (SSc)-associated ILD (SSc-ILD) therapy. Recently, it has been approved for the treatment of other progressive fibrosing ILDs [5,6,7,8,9,10]. Nintedanib is a potent oral inhibitor of the tyrosine kinase activity of several pro-angiogenic receptors: vascular endothelial growth factor receptors (VEGFR) 1-3, fibroblast growth factor receptors (FGFR) 1-3, and platelet-derived growth factor receptors (PDGFR) α and β [5,6,7,8,9,10].The most common drugs that have demonstrated major interactions with nintedanib are: dexamethasone, carbamazepine, rifampicin, phenobarbital, leflunomide, tripanavir, and phenytoin [5,6,7,8,9,10].

The most common adverse events (AE) of antifibrotic therapy involve the gastrointestinal tract [5,6,7,8,9,10]. The TOMORROW and INPULSIS trials showed that the AE most commonly associated with daily nintedanib at 300 mg was diarrhea, reported in 61.5% of cases [5,6,7,8,9,10]. In most patients, nintedanib-associated gastrointestinal problems can be managed by reducing the dose (200 mg/day), discontinuing treatment, and applying symptomatic treatments such as loperamide or similar [5,6,7,8,9,10]. The most common AE associated with pirfenidone in the CAPACITY and ASCEND studies was nausea, which appeared in 35.5% of patients [10,11,12,13,14,15]. Gastrointestinal toxicity associated with pirfenidone is managed by reducing the dose or interrupting treatment [10,11,12,13,14,15]. Photosensitivity and rash associated with pirfenidone appear mostly in the first months of treatment. This AE can be reduced by the use of photoprotective creams [10,11,12,13,14,15]. In addition, nintedanib and pirfenidone may cause an increase in liver enzymes. Dose adjustments made to manage AE do not reduce the effectiveness of nintedanib or pirfenidone in decreasing forced vital capacity [10,11,12,13,14,15]. Moreover, several studies have shown that efficacy and safety data in clinical practice are similar to those described in clinical trials [10,11,12,13,14,15].

## 3. Gastroesophageal Reflux Disease (GERD)

Gastroesophageal reflux disease (GERD) can be described as the symptoms complex (occurring at least twice a week) or complications due to the ascent of gastric contents into the esophagus and higher parts of the digestive system [8,9,10,11,12,13,14,15,16,17,18].

In the last few years, the prevalence of GERD has increased all over the world, including 27.8% of the general population of the USA and 25.8% of European citizens. The reasons for the apparent increase in the prevalence of GERD symptoms and erosive esophagitis are not clear [19,20]. Increased awareness of the disease and improved diagnostic techniques could play a role; there is evidence to suggest that eradicating *Helicobacter pylori* infection could provoke reflux esophagitis [21,22]. Moreover, a high body mass index has been associated with an increase in the risk of GERD and its complications [23].There is currently an epidemic of obesity in the United States as well as several European countries, which might explain part of the observed increase and contribute to additional future increases in GERD [21,22,23].

Patients with IPF, systemic sclerosis (SSc), asthma, and COPD are affected by GERD with higher prevalence than in the general population [24,25,26,27,28,29,30]. Several studies have supported the hypothesis that decreased pressures in the upper and lower esophageal sphincters and relaxation of the LES may contribute to microaspiration of small droplets of refluxate, which causes, over long periods of time, subclinical lung injury and fibroproliferative responses at the molecular and cellular levels, leading to pulmonary fibrosis [13,30,31,32,33,34,35,36,37,38,39].This hypothesis is also supported by the detection of gastric contents (acids and pepsin) in the bronchoalveolar lavage fluid (BALF) of patients with IPF [31,38,39,40].In particular, in SSc, a systemic autoimmune disease, the gastrointestinal involvement is present in up to 90 % of patients [25,26,27,28]. The most affected areas are the esophagus and the anorectal tract. Reflux, heartburn, and dysmotility are the leading causes of gastrointestinal discomfort.

Common symptoms of GERD are chest or epigastric pain, nausea, regurgitation, and bloating. We also mention cough, throat pain or burning, wheezing, and sleep anomalies among extraesophageal symptoms. GERD can be completely asymptomatic and coincidentally diagnosed. Therefore, an accurate (and timely) diagnosis of GERD remains difficult. There are a range of available diagnostic tests (e.g., an endoscopy to look for esophageal injury, 24-h ambulatory pH, etc.), although they all have important limitations (e.g., in reliability and the relationship between endoscopic results and patient symptoms) [25,26,27,28].

## 4. GERD-IPF Relationship

The number of newly diagnosed GERD patients included a steady increase in those with IPF in the last few years; we can also say the same for the prevalence of hiatal hernia, associated with erosive esophagitis [41,42,43,44]. Over the years, the association between GERD and IPF has stimulated trials of medical and surgical therapies; nevertheless, a causal relationship between GERD and IPF is not well defined yet. The fundamental question remains herein unanswered: *is GERD a consequence of IPF or is it a risk factor for it?* Some studies have hypothesized that GERD in IPF is more frequent because of potential confounding factors, such as advancing age and smoking [41,42,43,44,45,46,47,48].

Although knowledge of the pathogenesis of IPF remains incomprehensive, various individual genetic and epigenetic factors have been correlated with the development of fibrosis and potential contributions of some environmental exposures (e.g., GERD, smoking, infections, inhaled toxic material) have been presumed. In particular, several studies have hypothesized that GERD-associated microaspiration may lead to persistent inflammation impairing lung infrastructure, thereby possibly accelerating the progression of IPF. Repeated microinjury to alveolar epithelial tissues has been revealed as the first trigger of an aberrant repair process in which several lung cells develop abnormal behaviors that promote the fibrotic process [10,11,12,13,14,15,46,47,48]. In addition, IPF may increase intrathoracic pressure, which can worsen GERD [47,48].

Currently, antacid therapy (e.g., proton pump inhibitors and histamine-2-blocker receptor antagonists) for IPF patients with GERD symptoms has only a conditional recommendation according to IPF guidelines [2,15,38,49,50]. Some studies show that it slows disease progression in terms of lung function decline, improves survival, decreases the number and frequency of acute exacerbations of the disease, and prolongs transplant-free survival; however, there are post-hoc analyses which have demonstrated no benefit from antacid therapyover a placebo in patients following treatment with pirfenidone and Nintedanib [2,38,50,51,52,53,54,55,56,57].

Laparoscopic fundoplication is another treatment option for patients with IPF and GERD because theoretically it may help to control reflux and avoid aspiration, with the same effects mentioned above on the progression of the disease. [38,50,51,52,53,54,55].

## 5. Hiatal Hernia and IPF

Hiatus hernia (HH) is an alteration of the integrity of the LES, a frequent finding by both radiologists and gastroenterologists, which can be associated with episodes of microaspiration, LES pressure reduction, and an increased risk of esophagitis [58]. As for patients with IPF, it is common to find pathological reflux, although it is difficult to define its causal relationship with lung damage [59,60]. In addition, the presence of symptoms does not differ between those with and without reflux.

The prevalence of HH increases with both age—particularly the 6th decade—and obstructive airway physiology, like asthma. The risk factors of HH include increased abdominal pressure (obesity, pregnancy, etc.), coughing, and tobacco smoking [59,60,61].

A barium esophagram, esophageal endoscopy, and high-resolution CT can help to describe the anatomy of the esophagus in relationship to the diaphragm and find the location of the LOS [61,62,63,64,65,66]. Different studies reported the presence of HH in between 39% and 53% of IPF patients, according to the method of CT interpretation [42,62]. HH may increase the probability of IPF due to easier microaspiration in affected patients. Lee et al. [62] found measurable concentrations of pepsin in the BALF of IPF patients, suggesting that these episodes may be frequent in these patients (Figure 1).

It is fundamental to understand whether HH in IPF patients is a comorbidity, a consequence, or a driver of IPF (or all of these, or a complex combination). Probably, extensive fibrosis causes a reduction in lung compliance, shifting the diaphragmatic pillars and increasing the prevalence of HH in IPF patients [43]. CT fibrosis scores and respiratory function are similar in IPF patients with and without HH [43].

It is still not clear whether HH can be considered a predictor of mortality from respiratory causes in IPF patients [42]. Indeed, anti-acid therapy showed a slower decline in FVC compared to patients not receiving these drugs [38]. As HH increases the risk of recurrence and medical therapy failure, surgical correction could represent a potential therapeutic strategy in the future [67].

## 6. Therapeutic Strategies for GERD in IPF

The therapeutic options to address GERD in adults include: diet and lifestyle changes (such as weight loss for overweight subjects, tobacco and alcohol cessation, staying upright during and after meals, avoidance of potentially aggravating foods, etc.) [68,69]; medical treatments, some of which neutralize the gastric acid—namely antacids, H2RA, and PPIs [68,70,71]—while others act by alternative means, such as prokinetics, Baclofen, and sucralfate [68,72,73,74,75,76,77] and surgical and endoscopic approaches such as fundoplication [78,79,80,81,82], magnetic sphincter augmentation [83,84,85,86], Roux-en-Y gastric bypass [87], radiofrequency antireflux treatment (Stretta; Restech, Houston, TX, USA), and TIF (endogastric solutions) [88,89,90]. All details about the medical options are described in Table 1.

Focusing our attention on GERD treatment in patients with IPF, in 2015 an American preclinical study analyzed the effect of PPIs in mitigating lung injury and fibrosis in IPF, suppressing acid reflux, and arresting cytokines release. This study started from the evidence that PPI use is associated with increased longevity in IPF patients and analyzed esomeprazole action in reducing lung inflammation and fibrosis in rats, concluding that probably through GERD treatment it is possible to find a therapeutic effect on pulmonary fibrosis [35,91,92,93,94,95,96,97,98,99,100,101,102,103,104]. Of course, the limitation of these results is that they need to be confirmed in prospective clinical studies. Another study [92], published in July 2022, confirms how antireflux surgery in IPF patients with GERD is able to improve lung function. Nelkine et al. [93] also tried to evaluate new options of molecules, like the promising quercetin, a flavonoid antioxidant, which seems to have the most significant therapeutic effects for both GERD and IPF.

Nevertheless, at the same time, a systematic review [94] published in May 2022 showed that antacid medication and antireflux surgery do not clearly improve respiratory outcomes in patients with IPF; a cohort study [95] concluded the same, claiming that PPI use was not associated with lower mortality or hospitalization incidence in patients with IPF. It is obvious, then, that a clear therapeutic strategy to follow for IPF patients with GERD is far from being defined.

## 7. Narrative Review of Literature

We performed a review of the literature in PubMed^®^ in order to identify relevant full-text manuscripts treating the GERD-ILD relationship from different points of view. The search was performed using the following keywords or synonyms: ‘GERD ILD’, ‘GERD IPF’, and ‘GERD Idiopathic pulmonary fibrosis’, without limitation on the date of publication, up to the 12th of March 2022.

Records were selected only if they were strictly treating the specified topic and in particular those with a focus on pulmonary perspective and with a larger number (*n* > 20) of patients examined. We included reviews, meta-analyses, case studies, and retrospective studies, without any limitation on the language of the original text.

We selected 14 papers, of which 6 were original studies. Four were case–control studies; the two others were a retrospective study and a meta-analysis. We chose these six papers for their different designs, for the large number of examined patients (> 100 in the two case–control studies, > 1000 in the meta-analysis), and in order to show that, if the association of GERD-IPF is well known, clinical implications still need to be discussed. Table 2 contains the principal characteristics of the 14 included studies, comprising 4392 patients with IPF and 658 with GERD. Four case–control studies that investigated the strict relationship between GERD and IPF were included in this systematic literature review (Table 2). Gribbin et al. [97] found a significant association between exposure to GERD and IPF and the strongest association with the prescribing of ulcer drugs (OR 2.20). The same association was confirmed also by Gao et al. [32] (62% prevalence of GERD in IPF patients) and Savarino et al. [31] who observed 40 patients with IPF, 40 with non-IPF lung fibrosis, and 50 healthy subjects, finding an increased number of GERD episodes in patients with IPF who had significantly higher (*p* < 0.01) oesophageal acid exposure and risk of gastric aspiration. In particular, IPF patients had more bile acids (present in 61% of IPF patients compared to none of the healthy volunteers (*p* < 0.001)) and a more severe GERD—which can be a reason for lung damage and fibrotic evolution, the same evolution described by Baqir et al. [103]—with a 1.78 OR of having GERD in IPF cases compared with the population controls.

The 2019 meta-analysis, [48] which included 3206 subjects and triple the number of controls *(n* = add), identified an association between GERD and IPF (OR, 2.94 (95% CI, 1.95–4.42); homogeneity < 0.0001), although the potential for confounding exists, e.g., smoking. It is also remarkable that this meta-analysis was based exclusively on case–control studies, so the confidence in the estimate of association is low, as the authors themselves clarified.

The retrospective study conducted from the analysis of the IPF Australian Registries [101] showed that the control of GERD symptoms and a specific medical therapy were not able to change IPF patients’ outcomes, thus it did not advise the use of antiacid therapy with this aim but simply to control the GERD symptoms.

## 8. Discussion

The possible etiologic role of reflux in pulmonary fibrosis has been advanced on the basis of the results of several studies that have demonstrated the high frequency of GERD in the IPF population. A major challenge is that gastroesophageal reflux is frequently asymptomatic, but a diagnosis is typically based on the presence of common symptoms and therefore often inaccurate.

A major unresolved issue is to elucidate whether GERD is a direct driver of IPF or a simple consequence or innocent bystander, potentially due to a decreased pulmonary compliance during respiratory ventilation secondary to fibrosis. Manometric study of esophageal motor function in IPF patients has not demonstrated particular patterns of body dysmotility, and with only a few studies indicating a higher frequency of ineffective peristalsis [28,55,56]. Regarding the two esophageal sphincters, the LES has almost always a normal tone, while the UES (upper esophageal sphincter) is more frequently hypotonic [28,55,56]. Until now, various studies used different criteria for the diagnosis of GERD, but its prevalence can be really estimated only by means of MII-pH (Multichannel Intraluminal Impedance-pH testing). Impedance studies have shown that refluxes can frequently reach the proximal esophagus, explaining how this can subsequently involve the pulmonary tree. This proximal extension of GERD is often coupled with a slower acid clearance. Another aspect issued from MII-pH analysis, but not fully clarified, is the possible role of non-acid reflux and other components such as pepsin and biliary salts, repeatedly found in the BALF of patients with IPF and GERD. These elements are present only in patients with GERD detected by means of objective measurements and could play an important role in the pathogenesis of IPF.

From a therapeutic point of view, the role of proton pump inhibitor (PPI) in GERD was studied particularly in the last decade [28,50,55,56]. The latest international guidelines for IPF [104] are critical about using antisecretory therapy in these patients. Several studies reported that PPI can be of help in improving lung function, reducing the decline of pulmonary functional tests, diminishing the rate of exacerbation of the disease, and improving the disease prognosis. The evidence to date is, unfortunately, mostly of low quality and based on observational and retrospective studies; therefore, randomized trials comparing placebo and antiacid medication would add essential information to the topic and could perhaps define definitive recommendations [104]. PPIs may play a role in IPF beyond controlling GERD pH itself or microaspiration. An alternative and biologically plausible mechanism involves the downregulation of fibroinflammatory molecules, the inhibition of fibroblast proliferation, and the upregulation of cytoprotective mechanisms. Surgery represents the other treatment option for GERD in IPF and is little studied in this context. LARS (laparoscopic antireflux surgery) is almost always safe and, for selected patients, represents a good therapeutic option. Its postoperative complications are limited to gas-bloat syndrome, dysphagia, diarrhea, and recurrent heartburn, which typically improve 3-6 months after surgery, sometimes with the help of dietary modifications, pharmacologic therapies, and esophageal dilation [28,50,55,56]. Unfortunately, our current data far from confirm a superior outcome for patients when surgery is used in the management of IPF-GERD [28,50,55,56].

Regarding GERD diagnosis, esophageal MII-pH is the current gold standard. Two abstracts have shown the correlation between mean nocturnal baseline impedance (MNBI) and postreflux swallow-induced peristaltic wave (PSPW) values and changes in pulmonary function testing parameters over 1 year in IPF patients [28,50,55,56]. Low proximal and distal MNBI were positively correlated with a 1-year decline in FEV1 (*r*=0.50, *p*=0.03) and FVC. The authors concluded that low distal and proximal MNBI, as well as an abnormal PSPW, are predictors of a more severe decline in lung function over 1 year on pulmonary function tests among pre-lung transplant IPF patients [28,50,55,56]. Another tool emerging in the field of esophageal pathophysiology is the functional lumen imaging probe (FLIP) and its endoscopic subtype, EndoFLIP (Crospon Ltd., Galway, Ireland), for the assessment of the mechanical properties of the esophageal wall and both LES and UES compliance [28,50,55,56].

Another possible application of EndoFLIP could be the identification of those patients presenting EGJ (esophagogastric junction) dysfunction that could benefit from surgery. All these new diagnostic tools, in addition to a more diffuse and frequent use of MII-pH, should improve our ability to establish a more accurate cause–effect relationship between IPF and GERD and to assess the real effectiveness of both medical and surgical antireflux therapies [28,50,55,56].

## 9. Conclusions

The examined data strongly support an association between GERD and IPF. Therefore, all IPF patients should be objectively screened for GERD by pathophysiological studies, mainly using HRM (high-resolution manometry) and potentially MII-pH. Moreover, bronchoscopy with analysis of BALF by dosing pepsin and/or bile acids could represent a diagnostic option in order to increase diagnostic accuracy, but it is seldomly performed in routine clinical practice for IPF due to an increased risk of exacerbations; therefore, it cannot be considered in the diagnostic routine for GERD-IPF. Furthermore, the efficacy of PPI therapy, criticized in the latest IPF guidelines, should be evaluated in prospective and randomized clinical trials against placebo, aimed to assess lung function parameters and the clinical response of respiratory symptoms and to improve oesophageal pH and impedance values. Antireflux surgery, particularly LARS, appears to be an effective and safe option for these patients. In conclusion, our current uncertainty about the GERD-IPF relationship and, consequently, about its therapeutic management, is due to the poor knowledge of the real pathogenic mechanisms of the pulmonary damage in IPF (Table 3). Several questions need to be addressed, particularly in relation to the role of microaspiration in lung injury, remodeling, and progression to IPF. Moreover, there is a prominent need for high-quality, randomized, and controlled clinical studies to evaluate the correlation between GERD and IPF and to evaluate the efficacy of different treatment regimes.

## Figures and Tables

**Figure 1 pharmaceuticals-15-01033-f001:**
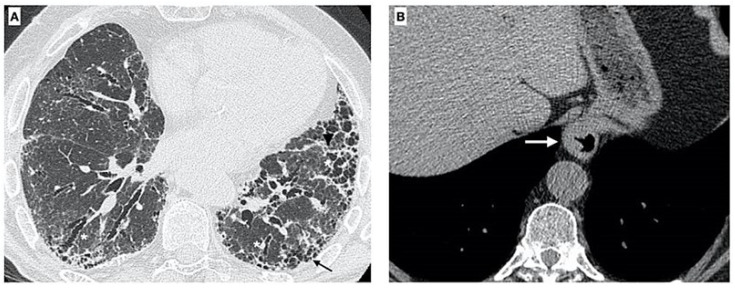
Axial chest CT images of a 73-year-old male with established IPF. (**A**) High-resolution image shows fibrotic changes due to the presence of diffuse irregular thickening of interlobular septa, traction bronchiectasis/bronchiolectasis, and honeycombing (black arrow); moreover, distortion of oblique fissure is visible. (**B**) CT also demonstrates the presence of a gastric (thoracic) hiatus hernia (white arrow).

**Table 1 pharmaceuticals-15-01033-t001:** Summary of antireflux drugs in IPF.

Medical Option	Effect
Antacids	Neutralize gastric acid and reduce acid delivery to the duodenum
Histamine-2 receptor antagonists (H2RAs) (e.g., cimetidine, famotidine, and nizatidine)	Inhibit acid secretion by blocking H2 receptors on the parietal cell
Proton pump inhibitors (PPIs) (e.g., omeprazole, lansoprazole, rabeprazole, pantoprazole, and esomeprazole)	Block acid secretion by irreversibly binding to and inhibiting the hydrogen-potassium ATPase pump that resides on the luminal surface of the parietal cell membrane
Prokinetics	Only in carefully selected patients who have GERD along with delayed gastric emptying and constipation
Baclofen	Inhibits the transmission of monosynaptic and polysynaptic reflexes at the spinal cord level, with resultant relief of muscle spasticity
Sucralfate	Prevents acute, chemically induced mucosal damage; heals chronic ulcers; stimulates angiogenesis and the formation of granulation tissue; binds to the injured tissue, thereby delivering growth factors and reducing access to pepsin and acid

**Table 2 pharmaceuticals-15-01033-t002:** Table of the studies which investigated the strict relationship between GERD and IPF, which were included in this systematic literature review.

Authors	Title of the Article	Design	Examined Patients	Results
Gribbin et al. (2008) [97]	Role of diabetes mellitus and gastro-oesophageal reflux in the etiology of idiopathic pulmonary fibrosis	Case–control	920 patients with IPF, 3593 control subjects	The study provides further evidence of an association between idiopathic pulmonary fibrosis and both diabetes mellitus and gastro-oesophageal reflux.
Bandeira et al. (2009) [37]	Prevalence of gastroesophageal reflux disease in patients with idiopathic pulmonary fibrosis	Prospective study	28 patients with IPF, 10 of them with GERD	The prevalence of GERD in the patients with IPF was high. However, the clinical and functional characteristics did not differ between the patients with GERD and those without.
Lee et al. (2011) [38]	Gastroesophageal Reflux Therapy is Associated with Longer Survival in Patients with Idiopathic Pulmonary Fibrosis	Retrospective cohort study	204 patients, 96 in treatment for GERD	The reported use of GERD medications is associated with decreased radiologic fibrosis and is an independent predictor of longer survival time in patients with IPF.
Soares et al. (2011) [33]	Interstitial Lung Disease and Gastroesophageal Reflux Disease: Key Role of Esophageal Function Tests in the Diagnosis and Treatment	Prospective study	44 patients with respiratory disease	(a) Pathologic distal GERD is present in more than two-thirds of patients with ILD, and abnormal proximal GERD is present in 20%; (b) typical reflux symptoms are a poor predictor of pathological reflux; and (c) esophageal function tests are essential for establishing the correct diagnosis and for the treatment of these patients.
Allaix et al. (2013) [98]	Idiopathic Pulmonary Fibrosis and Gastroesophageal Reflux. Implications for Treatment	Retrospective review of a prospectively set institutional review	22 patients with IPF and GERD, 80 with GERD	In patients with GERD and IPF (a) reflux is frequently silent, (b) with the exception of a weaker UES, the esophageal function is preserved, and (c) proximal reflux is more common, and in the supine position it is coupled with a slower acid clearance. Because these factors predisposing IPF patients to the risk of aspiration, antireflux surgery should be considered early after the diagnosis of IPF and GERD is established.
Savarino et al. (2013) [31]	Gastro-oesophageal reflux and gastric aspiration in idiopathic pulmonary fibrosis patients	Case–control	40 patients with IPF, 40 with non-IPF lung fibrosis, and 50 healthy patients	IPF patients had significantly higher (p,0.01) oesophageal acid exposure. IPF and non-IPF patients differed from healthy volunteers only in terms of mean LOS basal pressure and prevalence of hiatal hernia.
Gao et al. (2015) [32]	The prevalence of gastro-esophageal reflux disease and esophageal dysmotility in Chinese patients with idiopathic pulmonary fibrosis	Case–control	69 patients with IPF, 88 patients with GERD, 62 healthy patients	GERD prevalence in IPF was high, but symptoms alone were an unreliable predictor of reflux
Allaix et al. (2017) [99]	Gastroesophageal Reflux and Idiopathic Pulmonary Fibrosis	Review		Medical therapy with acid-reducing medications controls the production of acid and has some benefit. However, reflux and aspiration of weakly acidic or alkaline gastric contents can still occur.
Wang et al. (2018) [100]	Gastroesophageal Reflux Disease in Idiopathic Pulmonary Fibrosis: Uncertainties and Controversies	Review		GERD is highly prevalent in IPF and may play a role in its pathogenesis and progression through microaspirations.
Methot et al. (2019) [48]	Meta-analysis of Gastroesophageal Reflux Disease and Idiopathic Pulmonary Fibrosis	Meta-analysis	3200 patients with IPF, 9368 control subjects	GERD and IPF may be related, but this association is most likely confounded, especially by smoking. Our confidence in the estimate of association is low because it is exclusively from case–control studies.
Jo et al. (2019) [101]	Gastroesophageal reflux and antacid therapy in IPF: analysis from the Australia IPF Registry	Retrospective cohort study	587 patients, 384 in treatment for GERD	Neither the use of antacid therapy nor the presence of GORD symptoms affects longer-term outcomes in IPF patients. This contributes to the increasing evidence that antacid therapy may not be beneficial in IPF patients.
Ghisa et al. (2019) [102]	Idiopathic pulmonary fibrosis and GERD: links and risks	Review		All this uncertainty about the GERD-IPF relationship and, consequently, about its therapeutic management, is due to the scant knowledge of the real pathogenic mechanisms of the pulmonary damage in IPF.
Nelkine et al. (2020) [93]	Role of antioxidants in the treatment of gastroesophageal reflux disease-associated idiopathic pulmonary fibrosis	Review		There is a connection between GERD and IPF. As current treatment options are still inadequate to improve the condition and increase the survival rate of IPF patients, alternative treatment options are crucial. Based on the reviewed scientific evidence, antioxidant supplementation could complement standard IPF treatment, certainly in GERD-associated IPF.
Baqir et al. (2021) [103]	Idiopathic Pulmonary Fibrosis and Gastroesophageal Reflux Disease: A Population-Based, Case–Control Study	Case–control	113 patients with IPF, 226 patients with ILD non-IPF, 226 patients with case–controls	GERD may be an important contributor to the development of lung fibrosis. Thus, it should be investigated and addressed adequately when detected in patients with IPF and patients with non-IPF ILD.

**Table 3 pharmaceuticals-15-01033-t003:** Clinical recommendations and future directions for GERD-IPF association.

Clinical Recommendations	Future Directions
Screening for GERD in all IPF patients with HRM and potentially MII-pH	Focus on mechanism of pulmonary damage in IPF
Antireflux surgery with LARS	Prospective and randomized clinical trials for GERD treatments against placebo

## Data Availability

Data sharing not applicable.

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
