# Peer review of "Gastroesophageal Reflux Disease in Idiopathic Pulmonary Fibrosis: Viewer or Actor? To Treat or Not to Treat?"

_pharmaceuticals, 2022, doi:10.3390/ph15081033_

Round 1

Reviewer 1 Report

I have read the article by Ruaro et al. with great interest. This review focuses on the association between GERD and IPF. This relationship is well known, yet the clinical implications are limited. The review updates us on this correlation. The information on the cause-consequence relationship between the two diseases is frequently repeated, while other important information (ie. why would reflux cause IPF) is not fully discussed.

Comments:

·       The abstract submitted to the reviewer platform is totally different from the one in the manuscript. Please, clarify which is the correct version.

·       2. Antifibrotics drugs for IPF and gastrointestinal adverse events. You mention a few potential drug interactions with antifibrotics. Please, provide further explanation if they interact with any of the anti-reflux medications.

·       4. Relationship GERD-IPF. Please, provide mechanistic explanation how would reflux cause IPF. Which inflammatory/fibrotic pathway would it induce?

·       Figure 1. The slices are not at the same level. Is it the same patient? Figure 1a shows IPF without any indication for hiatus hernia, Figure 1b shows hiatus hernia without indication for IPF. Please, provide an image which provides both modalities.

·       6. Therapeutic strategies for GERD in IPF. Please, provide a Table summarising the effect of anti-reflux drugs in IPF.

·       Conclusions. Bronchoscopy with BAL is not routinely recommended in IPF as it is associated with risk for exacerbation. Therefore, I disagree with the statement regarding BALF pepsin/bile acid measurements.

·       Conclusion. Please provide clinical recommendations and future research needs in two tables so that the information is better visualised for the readers.

·       Sometimes you write GER, GERD or GORD. Please, use the same terminology throughout the manuscript.

Author Response

I have read the article by Ruaro et al. with great interest. This review focuses on the association between GERD and IPF. This relationship is well known, yet the clinical implications are limited. The review updates us on this correlation. The information on the cause-consequence relationship between the two diseases is frequently repeated, while other important information (ie. why would reflux cause IPF) is not fully discussed.

R: We would like to thank the reviewer for the useful comments that help us to ameliorate our manuscript.

Comments:

  •      The abstract submitted to the reviewer platform is totally different from the one in the manuscript. Please, clarify which is the correct version.

R: We would like to thank the reviewer for this observation. I apologize for my mistake, the correct version of the abstract is reported in the manuscript.

  • 2. Antifibrotics drugs for IPF and gastrointestinal adverse events. You mention a few potential drug interactions with antifibrotics. Please, provide further explanation if they interact with any of the anti-reflux medications.

R: In agreement with the reviewer’s comment, we add these paragraphs in the manuscript: “The most common adverse reactions are observed at the gastrointestinal, skin and liver level. It is worth highlighting that polymedicated patients should be closely monitored, as pirfenidone metabolism can be influenced in these patients by inhibition or induc-tion of liver enzyme systems such as cytochrome P450 1A2 (CYP1A2), CYP3A4 and P-glycoprotein (P-gp) 10−15. The most common drugs that demonstrated major interac-tions with pirfenidone are: amiodarone, enoxacin, leflunomide, fluvoxamine, ami-nolevulinic acid, mipomersen, mibefradil, rucaparib and teriflunomide” and “Nintedanib is a potent oral inhibitor of the tyrosine-kinase activity of several pro-angiogenic receptors: vascular endothelial growth factor receptors (VEGFR) 1-3, fibroblast growth factor receptors (FGFR) 1-3 and platelet-derived growth factor re-ceptors (PDGFR) α and β 10−15. The most common drugs that demonstrated major in-teractions with nintedanib are: dexamethasone, carbamazepine, rifampicin, pheno-barbital, leflunomide, tripanavir adn phenytoin”.

  • 4. Relationship GERD-IPF. Please, provide mechanistic explanation how would reflux cause IPF. Which inflammatory/fibrotic pathway would it induce?

R: Thank you, in agreement with the reviewer’s comment we insert this paragraph in the paper:

Although knowledge of the pathogenesis of IPF remains incomprehensive, various individual genetic and epigenetic factors have been correlated with the development of fibrosis and potential contributions of some environmental (e.g. GERD, smoking, in-fections, inhaled toxic material) exposures have been presumed. In particular, several studies hypothesize that GERD-associated microaspiration may lead to persistent in-flammation impairing lung infrastructure, thereby possibly accelerating the progres-sion of IPF. Repeated microinjury to alveolar epithelial tissues has been revealed as the first trigger of an aberrant repair process in which several lung cells develop abnormal behaviors that promote the fibrotic process. 10−15,47,48 In addition, IPF may increase in-trathoracic pressure, that can worsen GERD. 47,48 

  • Figure 1. The slices are not at the same level. Is it the same patient? Figure 1a shows IPF without any indication for hiatus hernia, Figure 1b shows hiatus hernia without indication for IPF. Please, provide an image which provides both modalities.

R: I would like to thank you the reviewer for this comments. The two CT images refer to the same patient. The reason why we chose two different CT levels was to show parenchymal findings in the first image and the gastric hernia in the second one with greater precision. If we showed the two findings in a single image, we would have to use a different reconstruction filter dedicated to mediastinum study, risking not to clearly show the lung parenchyma and consequently fibrosis.

  • 6. Therapeutic strategies for GERD in IPF. Please, provide a Table summarising the effect of anti-reflux drugs in IPF.

R: Thank you for your comment. We added table 1 in the manuscript in order to clarify different option of medical treatment for GERD.

  • Conclusions. Bronchoscopy with BAL is not routinely recommended in IPF as it is associated with risk for exacerbation. Therefore, I disagree with the statement regarding BALF pepsin/bile acid measurements.

R: We modified conclusion section about BALF use in diagnostic routine. We specified that it represents a diagnostic option, underlying the risk of exacerbations that it has as you correctly suggested. We add this paragraph in the text: “9. Conclusions. Examined data strongly supports an association between GERD and IPF. There-fore, all IPF patients should be objectively screened for GERD by pathophysiological studies, mainly using HRM (High Resolution Manometry) and also potentially MII-pH. Also, bronchoscopy with analysis of BALF by dosing pepsin and/or bile acids could represent a diagnostic option in order to increase the diagnostic accuracy but is seldomly performed in routine clinical practice for IPF due to an increased risk of exacerbations, so it can not be considered in diagnostic routine for GERD-IPF. Furthermore, the efficacy of PPI therapy, criticized in the latest IPF guidelines, should be evaluated in prospective and randomized clinical trials against placebo aimed to assess lung function parameters and clinical response of respiratory symptoms and to improve oesophageal pH and impedance values. Anti-reflux surgery, particularly LARS, appears to be an effective and safe option for these patients. In conclusion, our current uncertainty about GERD-IPF relationship and, consequently, about its therapeutic management, is due to the poor knowledge of the real pathogenic mechanisms of the pulmonary damage in IPF (Table 3)”.

  • Conclusion. Please provide clinical recommendations and future research needs in two tables so that the information is better visualised for the readers.

R: In agreement with the reviewer’s observation, we added table 3 with two different section for clinical recommendations and future directions which emerge from this review.

  • Sometimes you write GER, GERD or GORD. Please, use the same terminology throughout the manuscript.

R: In accordance with the reviewer's comment, we replaced GER and GOR with GERD for a clearer reading

Reviewer 2 Report

Appreciate the opportunity to review this work. I believe is well written and brings value for the journal readers. I have a few comments that can be address and hopefully improve this publication. 

1.- a better detail of why publications were excluded as well as how GERD and IPF were define in the selected studies. trying to make sure there was some form of consistency between the review studies. if there is an issue with space limitation, at least a comment on the methods use to evaluate the data on the papers review for this publication

2.- the conclusion section starts with the statement Our data. I think it will be better to use a more general term. 

3.- the conclusion also suggest that all patients should be screened for GERD with different modalities, including performing a bronchoscopy. I suggest the authors to be careful with these recommendations, specially since the authors recognize that the effect of treating GERD might not have any effect over the course of IPF. 

Author Response

Appreciate the opportunity to review this work. I believe is well written and brings value for the journal readers. I have a few comments that can be address and hopefully improve this publication. 

R: Thank you very much for all your comments.

1.- a better detail of why publications were excluded as well as how GERD and IPF were define in the selected studies. trying to make sure there was some form of consistency between the review studies. if there is an issue with space limitation, at least a comment on the methods use to evaluate the data on the papers review for this publication

R: In agreement with the reviewer’s comment, we added other explanation about how we decided which papers use in our review: We selected 14 papers and in particular 6 were original studies. Four were case-control studies, the two others were a retrospective study and a metanalysis. We chose these 6 papers for their different design, for the large number of examined patients (> 100 in the two case-control studies, > 1000 in the methanalysis) and in order to show that if the association GERD-IPF is well knows, clinical implications still need to be discussed. Table 2 contains the principal characteristics of the 14 included studies, comprising 4392 patients with IPF and 658 with GERD. Four case control-studies which investigated the strict relationship between GERD and IPF, were included in this systematic literature review (Table 2). 

2.- the conclusion section starts with the statement Our data. I think it will be better to use a more general term. 

R: Thank you, we replaced "Our data" at the beginning of conclusion section with a more general "Examined data" as correctly suggested. 

3.- the conclusion also suggest that all patients should be screened for GERD with different modalities, including performing a bronchoscopy. I suggest the authors to be careful with these recommendations, specially since the authors recognize that the effect of treating GERD might not have any effect over the course of IPF. 

R: Thank you, we modified conclusion section about BALF use in diagnostic routine. We specified that it represents a diagnostic option, underlying the risk of exacerbations that it has, as suggested: “Also, bronchoscopy with analysis of BALF by dosing pepsin and/or bile acids could represent a diagnostic option in order to increase the diagnostic accuracy but is seldomly performed in routine clinical practice for IPF due to an increased risk of exacerbations, so it can not be considered in diagnostic routine for GERD-IPF”.

Reviewer 3 Report

In a recent update of guidelines about IPF (American Journal of Respiratory and Critical Care Medicine Volume 205 Number 9 | May 1 2022), conditional recommendation against antacid medication,was voted by 86% of committee’s participants; and strong recommendation against antacid medication,by 7%.

"The predominance of existing evidence is observational and,therefore, susceptible to bias due to unmeasured confounders. Randomized trials comparing the effects of antacid medication and placebo on respiratory outcomes in patients with IPF would be a valuable addition to the field, potentially enabling definitive recommendations"

Please include this study in the analisis of relults, discussion and conclusions

Author Response

In a recent update of guidelines about IPF (American Journal of Respiratory and Critical Care Medicine Volume 205 Number 9 | May 1 2022), conditional recommendation against antacid medication,was voted by 86% of committee’s participants; and strong recommendation against antacid medication,by 7%.

"The predominance of existing evidence is observational and,therefore, susceptible to bias due to unmeasured confounders. Randomized trials comparing the effects of antacid medication and placebo on respiratory outcomes in patients with IPF would be a valuable addition to the field, potentially enabling definitive recommendations"

Please include this study in the analisis of relults, discussion and conclusions

R: Thank you for your comment. The suggestion was valuable, as it allows the latest evidence available in the literature to be added to discussion and conclusion sections of the review, also providing precise advice on how to proceed in the research related to GERD-IPF association.

Round 2

Reviewer 1 Report

I am happy with the changes and suggest acceptance.

Reviewer 3 Report

Modifications performed are fine